# In Vitro Antioxidant and Anti-Glycation Activity of Resveratrol and Its Novel Triester with Trolox

**DOI:** 10.3390/antiox10010012

**Published:** 2020-12-24

**Authors:** Sebastiano Intagliata, Angelo Spadaro, Miriam Lorenti, Annamaria Panico, Edy A. Siciliano, Sabrina Barbagallo, Benito Macaluso, Shyam H. Kamble, Maria N. Modica, Lucia Montenegro

**Affiliations:** 1Department of Drug and Health Sciences, University of Catania, 95125 Catania, Italy; s.intagliata@unict.it (S.I.); angelo.spadaro@unict.it (A.S.); miriam.lorenti@gmail.com (M.L.); panico@unict.it (A.P.); edysiciliano@hotmail.it (E.A.S.); sabribar94@gmail.com (S.B.); benitomac@outlook.it (B.M.); 2Department of Pharmaceutics, College of Pharmacy, University of Florida, Gainesville, FL 32610, USA; kamble.shyam@cop.ufl.edu

**Keywords:** resveratrol, resveratrol derivative, anti-glycation, antioxidant, trolox, chemical stability

## Abstract

Resveratrol (RSV) is well known for its many beneficial activities, but its unfavorable physicochemical properties impair its effectiveness after systemic and topical administration; thus, several strategies have been investigated to improve RSV efficacy. With this aim, in this work, we synthesized a novel RSV triester with trolox, an analogue of vitamin E with strong antioxidant activity. The new RSV derivative (RSVTR) was assayed in vitro to evaluate its antioxidant and anti-glycation activity compared to RSV. RSVTR chemical stability was assessed at pH 2.0, 6.8, and 7.2 and different storage temperatures (5 °C, 22 °C, and 37 °C). An influence of pH stronger than that of temperature on RSVTR half-life values was pointed out, and RSVTR greatest stability was observed at pH 7.2 and 5 °C. RSVTR showed a lower antioxidant ability compared to RSV (determined by the oxygen radical absorbance capacity assay) while its anti-glycation activity (evaluated using the Maillard reaction) was significantly greater than that of RSV. The improved ability to inhibit the glycation process was attributed to a better interaction of RSVTR with albumin owing to its increased topological polar surface area value and H-bond acceptor number compared to RSV. Therefore, RSVTR could be regarded as a promising anti-glycation agent worthy of further investigations.

## 1. Introduction

Resveratrol (*trans*-resveratrol, RSV) is a natural substance occurring in some plants, particularly grapevine, peanuts, Japanese knotweed, and foods, such as wine, Itadori tea, soy, and red fruits [1,2,3,4]. RSV has been the subject of an extensive investigation by the academic community and industries due to a significant number of pharmacological properties in humans, such as antioxidant, anticancer, and anti-inflammatory activity [5,6,7,8,9,10], and for its ability to act as a protective agent on cardiovascular diseases and metabolic syndrome [11,12]. Unfortunately, RSV effectiveness is impaired by its unfavorable chemical–physical properties, such as low water solubility and susceptibility to oxidation reaction after oxygen and light exposure, limiting RSV bioavailability [13,14].

Recently, various approaches have been proposed to improve RSV biological activities after systemic and topical administration; these strategies include RSV incorporation into different types of nanocarriers and synthesis of prodrugs or RSV derivatives with improved chemical–physical characteristics compared to the parent drug [15,16,17,18,19,20,21,22,23]. In particular, several studies suggest the feasibility of improving RSV effectiveness by synthesizing derivatives in which molecules that might exert a synergic effect are linked to RSV (i.e., RSV hybrids and codrugs) [24,25]. In fact, hybrids and bifunctional ligands have drawn much attention from medicinal chemists and quickly became popular in the drug design and development process [26,27,28,29].

In this work, we synthesized and characterized a novel RSV triester with trolox, a water-soluble analogue of vitamin E extensively used in biological assays as a reference standard for its strong antioxidant activity. Esterifying RSV with trolox could result in a derivative with improved ability to counteract free radicals, thus limiting processes, such as glycation, in which oxidative mechanisms are involved.

Glycation is a non-enzymatic reaction between proteins and reducing sugars, leading to the formation of advanced glycation end products (AGEs), whose accumulation causes cell and tissue damage, as a result of lipid peroxidation, endothelial dysfunction, changes in protein structure, and stimulation of inappropriate cellular activity [30,31]. Recent reports suggest that increased glycation is associated with a higher prevalence of diabetes and related complications [32,33], myasthenia gravis [34], and neurodegenerative diseases [35]. Therefore, anti-glycation agents have been considered promising tools to cope with human aging and the development of several diseases. Increasingly evidence indicates that compounds of natural origin, especially polyphenols, have anti-glycation properties. In particular, several in vitro and in vivo studies highlighted that RSV could be useful in the treatment of diseases associated with increased glycation. A significant decrease in urine albumin and creatinine and an increase in serum antioxidant enzymes were observed in patients with type II diabetes and diabetic nephropathy who received RSV at the dose of 500 mg per day [36]. Furthermore, RSV is considered a prospective therapeutic agent against diabetic ototoxicity [37], renal fibrosis [38], the progression of cataracts [39], and bone density loss in patients with type 2 diabetes [40]. According to Liu et al. [41], RSV is able to inhibit AGEs-induced NO and PGE_2_ production in porcine chondrocyte cell cultures, thus effectively attenuating AGEs-induced cartilage damage. The authors concluded that RSV could be a promising therapeutic agent in treating osteoarthritis and other AGE-mediated diseases. Additionally, RSV showed anti-glycation activity against the harmful effect of AGEs on mouse oocytes [42] and dendritic cells obtained from peripheral blood mononuclear cells [43].

Therefore, in this work, the newly synthesized resveratrol derivative (RSVTR, Figure 1) was assayed in vitro to determine its antioxidant activity and its ability to inhibit AGEs compared to its parent drug RSV.

## 2. Materials and Methods

### 2.1. Materials

The following chemicals and solvents used to synthesize and characterize RSVTR were reagent grade and were bought from Sigma-Aldrich S.r.l. (Milan, Italy): resveratrol (RSV), trolox, *N*-(3-dimethylaminopropyl)-*N*′-ethylcarbodiimide hydrochloride (EDC. HCl), 4-dimethylaminopyridine (DMAP), dry dichloromethane (DCM), cyclohexane, ethyl acetate (EtOAc), hydrochloric acid (HCl), and DMSO-*d*_6_ used for NMR analysis.

Moreover, 2,2′-Azobis(2-methylpropionamidine) dihydrochloride (AAPH), fluorescein (FL), amino-guanidine carbonate (AMG), bovine serum albumin (BSA), D-fructose, sodium azide, triethylamine, and trifluoroacetic acid were purchased from Sigma-Aldrich S.r.l. (Milan, Italy). Acetonitrile and water of HPLC grade were obtained from Merck (Darmstadt, Germany).

### 2.2. Chemistry

The synthetic route to obtain RSVTR is reported in Figure 2. Commercially available RSV and trolox reacted in a 1:3 molar ratio, respectively. The reaction was carried out in the presence of EDC. HCl, a catalytic amount of 4-DMAP, and dry DCM as solvent under nitrogen atmosphere. The only compound isolated was the triester derivative of RSV with trolox, independently of the molecular reagent ratio.

RSVTR was purified by flash column chromatography using Merck silica gel (0.040–0.063 mm). The infrared spectrum was recorded on a Perkin Elmer series FTIR 1600 spectrometer (Milan, Italy) using a disk containing a pressed mixture of RSVTR and KBr powder. Signal intensity was characterized as s (strong), m (medium), and w (weak). ^1^H NMR spectrum was recorded on a Varian Unity INOVA 200 spectrometer (200 MHz) (Milan, Italy) using DMSO-*d*_6_ as a solvent. Chemical shifts are given in δ values (ppm), using tetramethylsilane as the internal standard; coupling constants (*J*) are given in Hz. Signal multiplicities are characterized as s (singlet), d (doublet), t (triplet), and m (multiplet). Reaction progress and purity of the synthesized compound were checked on thin layer chromatography (TLC, an aluminum sheet coated with silica gel 60 F254, Merck, Darmstadt, Germany) using cyclohexane/EtOAc (7:3, *v*:*v*) as eluents and spots were visualized by UV light at 254 and 366 nm as the wavelength.

### 2.3. Synthesis of 1,2,4-Benzenetricarboxylic Acid, 1,2,4-tris [2-[3,4-dihydro-2,5,7,8-tetramethyl-6-(phenylmethoxy)-2H-1-benzopyran-2-yl]ethyl] Ester (RSVTR)

A mixture of RSV (0.15 g, 0.657 mmol), trolox (0.493 g, 1.97 mmol), and 4-DMAP (0.008, 0.065 mmol) in dry DCM (10 mL) was prepared. EDC. HCl (0.306 g, 1.97 mmol) was added to the obtained suspension at 0 °C, under stirring and nitrogen atmosphere. Then, the mixture was stirred for 8 h at room temperature. The suspension, at the end of reaction time, was filtered under reduced pressure and the solution was washed with an aqueous NaHCO_3_ 5% solution (2 × 20 mL) and saturated H_2_O (1 × 20 mL). The organic layer was dried over anhydrous sodium sulfate and evaporated to dryness under reduced pressure to obtain an white crude product, which was purified by flash column chromatography using cyclohexane/ethyl acetate (7:3, *v*/*v*) as an eluent. Using evaporation in vacuo to dryness, the solvent of the collected homogeneous fractions, RSVTR was obtained as a white pure solid (0.62, 20.5%); mp 113.0-116.0. IR (KBr, cm^−1^, selected lines): 2929 (m), 1760 (s, broad bend), 1611 (w), 1506 (m), 1455 (s), 1462 (s), 1374 (w), 1338 (w), 1260 (s), 1089 (s), 964 (w), 676 (s). ^1^H NMR (DMSO-*d*_6_) d 1.71 (s, 9H, CH_3_CCO), 1.95–2.15 (m, 27H + 6H, 9CH_3_, 3CH_2_), 2.40–2.78 (m, 6H, 3CH_2_), 6.70–6.80 (m, 1H, ArH), 7.03 (d, *J* = 8.6 Hz, 2H, ArH), 7.18 (dd, *J* = 16.8, 2Hz, 2H + 2H, ArH, CH=CH), 7.54 (s, 3H, 3OH, exchanges with D_2_O), 7.61 (d, *J* = 8.8 Hz, 2H, ArH).

### 2.4. Chemical Stability

The chemical hydrolysis of RSVTR was studied at pH 2.0 (0.2 N HCI), 6.8 and 7.2 (phosphate buffer solutions). All solutions containing the same amount of RSVTR (1.18 µM) were kept in the dark at a constant temperature of 37 °C, 22 °C, and 5 °C. At appropriate time intervals, chosen to obtain enough data to pinpoint the stability profile (0, 2, 4, 8, 24, 48, 72, 96, 168, 696 h at pH 7.2; 0, 2, 4, 8, 24, 48, 72, 96, 192, 528, 768 h at pH 6.8; 0, 2, 4, 8, 24, 48, 72, 96, 528 h at pH 2.0), samples were withdrawn and analyzed immediately by HPLC for the residual RSVTR.

Pseudo-first-order rate constants were determined from the slope of the linear portion of the graph obtained, plotting the logarithm of the residual amount of RSVTR against time. RSVTR half-life (t_1/2_) was calculated according to Equation (1):
t_1/2_ = ln 0.5/k (1)
where k is the pseudo-first-order rate constant.

### 2.5. HPLC Analysis

HPLC analyses were performed on an Agilent 1260 Infinity II chromatographic system (Agilent Technologies) equipped with a ChemStation OpenLab software (M8307AA), a quaternary pump (G7111B), a diode array detector (DAD, G7111B), a manual sample injector (G1328C) with a 20 µL loop and a thermostated column compartment (G1316A). Chromatographic separations were performed using Eurosphere II 100-3 C18A (150 × 4.6 mm) column with integrated precolumn Knauer (LabService Analytica, Anzola Emilia, Italy).

RSVTR was analyzed using an isocratic binary mobile phase consisting of 5% of acetonitrile and 95% of triethylamine 0.3% (*v*/*v*) in water (pH adjusted to 3.0 with trifluoroacetic acid). The flow rate was set at 1.0 mL/min and the column was maintained at 24 °C throughout the analysis. Chromatograms were acquired at 290 nm after analyzing RSVTR UV spectra in the range 200–400 nm and the retention time of RSVTR was approximately 5.40 min. The analytical method was validated according to ICH guidelines [44].

Identification of the RSVTR ester was performed by HPLC–DAD analysis by comparing the retention times and the UV spectra of the analyzed samples with those of authentic reference standards. In addition, peak-purity tests, evaluated by OpenLab software (Agilent Technologies) using photodiode-array detector spectra, were utilized to demonstrate the absence of coeluting peaks. The peak purity indexes were found to be >999 with respect to a theoretical value of 1000. This fact demonstrated that degradation products and buffer constituents did not interfere with the analyte peak [45].

A stock solution of RSVTR (1200.0 μg/mL) was obtained by dissolving an appropriate amount of standard compound in methanol. Working standard solutions of the analyte were daily prepared by adequate dilution with eluent phase of the calculated amount of the stock solution.

Quantification of RSVTR was performed using six-point external calibration curve (37.5–1200 µg/mL). Linear regression was performed using OpenLab software (Agilent Technologies), to calculate slope, intercept and correlation coefficient (R^2^). The equation of the calibration curve was y = 19.432626x + 25.211692 (x = µg/mL, y = area). The plot was linear in the concentration range tested with a correlation coefficient of 0.9999.

The limits of detection (LOD) and quantification (LOQ) were calculated using LOD = 3.3·(SD/S) and LOQ = 10·(SD/S) formulas, respectively, where S is the slope and the SD is the residual standard deviation of calibration curves. The LOD and LOQ were 2.91 and 8.82 µg/mL, respectively.

The intra- and inter-day accuracy and precision were determined at three concentration levels of RSVTR (37.5, 600, and 1200 µg/mL) on the same day and on three different days, respectively. Intra- and inter-day precision were reported as relative standard deviation (RSD%), with an acceptability limit set at ≤15%. Accuracy was calculated by comparing the mean assay results with the nominal concentrations, with the acceptability limit set at ± 15%. The acceptability criterions recommended by the ICH guidelines were met for all experimental data, with intra- and inter-day RSD % not exceeding 4%, and accuracy values in the range 98.67–101.61%.

### 2.6. Oxygen Radical Absorbance Capacity (ORAC) Assay

To evaluate the scavenging activity of RSV and RSVTR, the peroxyl radical (ROO•) was generated by thermo-decomposition of 2,2 azobis (2-aminopropane) dihydrochloride (AAPH) (100 mM). The ORAC assay determined the fluorescence decrease of fluorescein (10 nM) after its oxidation in the presence of AAPH and the investigated samples (100 μg/mL) properly diluted, as previously reported [46,47]. The assays were carried out at 37 °C, pH 7.0 recording fluorescence through a Wallac Victor 1420 Multilabel Counters fluorimeter (PerkinElmer, Waltham, MA, USA) (excitation λ = 540 nm, emission λ = 570 nm). Trolox (12.5 μM) and phosphate buffer were used as control standard and blank, respectively. The area under the quenching curve of fluorescein was used to calculate ORAC value of the investigated compounds using Origin^®^7 (OriginLab Corporation, Northampton, USA). ORAC Units, expressed as trolox micromole per microgram of sample (μmol/μg), were obtained according to Equation (2):
ORAC value (μmol/μg) = K(S sample − S blank)/(S Trolox − S blank) (2)
where K is the sample dilution factor, S the area under the fluorescence decay curve of sample, trolox or blank. Each experiment was performed in triplicate and data were expressed as mean ± SD

### 2.7. Anti-Glycation Activity

RSV and RSVTR ability to prevent AGEs formation was assessed using the method reported by Derbrè et al. [48], producing AGEs through Maillard reaction. Protein model bovine serum albumin (BSA) (10 mg/mL) incubated with D-fructose (0.5 M) in phosphate buffer 50 mM, pH 7.4 and NaN3 0.02% was used as positive control. BSA alone was the negative control while aminoguanidine (AMG) (400 μg/mL) was used as a reference. Final glycated BSA solutions (300 μL), alone and with samples (400 μg/mL), were incubated at 37 °C in 96-well microtiter closed with their silicon lids for 7 days. The inhibition of fluorescence was determined using a Wallac Victor 1420 Multilabel Counters fluorimeter (PerkinElmer, Waltham, MA, USA) (λexc 370 nm; λem 440 nm). Relative fluorescence units (RFU) were recorded and the percentage of AGEs formation inhibition was calculated as follows (Equation (3)):
% of inhibition = {1 − (RFU sample/RFU Positive control)} × 100 (3)

Each experiment was performed in triplicate and data were expressed as mean ± SD Statistical analysis of the results was performed using Student’s t-test (*p* < 0.05).

## 3. Results and Discussion

RSV, trolox, and RSVTR physicochemical properties were calculated with the MarvinSketch program and are listed in Table 1. The obtained data show that RSV esterification with trolox led to a decrease of melting point and water solubility compared to starting products and a high increase of calculated LogP (cLogP) and topological polar surface area (TPSA) values. Moreover, the complex structure of RSVTR provided an increase of H-bond acceptor (HBA) and rotatable bonds (RBN). TPSA is defined as the sum of all polar atoms at the surfaces of a molecule. The increase of this structural parameter and the increase of HBA could positively influence the interaction properties with human serum albumin (HAS). It has been demonstrated that in the case of bufadienolides (therapeutic agents with anticancer properties), their affinity for human serum albumin (HSA) is enhanced by increasing their TPSA values, indicating that drugs with a greater number of polar groups are bound more tightly to HSA compared to drugs with a lower number of polar groups. Moreover, a correlation between HBA numbers and the HSA drugs affinity has been reported, thus an increase of H-bond acceptor numbers enhanced the binding affinities for HAS [49].

RSVTR chemical stability was studied at pH 2.0, 6.8, and 7.2 to simulate the gastric, the intestinal and the plasmatic environments, respectively. As temperature may strongly affect drug stability, RSVTR chemical hydrolysis was determined at three different storage temperatures, namely 5 °C, 22 °C, and 37 °C, for each selected pH value. The percentage of residual RSVTR vs. time in the different experimental conditions previously mentioned is depicted in Figure 3, Figure 4 and Figure 5.

Plotting the percentage of the logarithm of residual RSV triester against time, RSVTR half-life for each set of experimental conditions was calculated and listed in Table 2.

As expected, the higher the temperature, the lower RSVTR chemical stability at the same pH value. A different trend was observed comparing RSVTR half-life at the same temperature but different pH values. As shown in Table 2, at 5 °C RSVTR half-life decreased by reducing the pH value of the medium while at 22 °C and 37 °C a better chemical stability was observed at pH 6.8. Plotting RSVTR half-life vs. temperature of storage at each pH value, an inverse relationship was observed (Figure 6). Although few data points did not allow performing a significant correlation analysis, the different slope of each set of data suggests an influence of pH stronger than that of temperature on RSVTR chemical stability. From RSVTR half-life values, a better chemical stability of this RSV derivative could be expected in the intestinal environment rather than at gastric or plasmatic level. Therefore, an oral administration of RSVTR could be customized by designing gastro-resistant formulations that release the active ingredient in the intestine. Additionally, RSVTR half-life at 37 °C and pH 7.2 could be regarded as long enough to allow RSVTR interaction with its target.

The ORAC assay confirmed the potent antioxidant activity of RSV already reported in the literature [50,51], while the RSV esterification with trolox to give RSVTR resulted in a lower antioxidant capacity (Figure 7). As reported in the literature [46], the number of hydroxyl groups in polyphenols strongly affects the antioxidant and anti-glycation activity of such compounds. In this work, we synthesized a derivative of RSV (a polyphenol) esterifying three hydroxyl groups of RSV with trolox. The lower antioxidant activity of RSVTR compared to RSV could be likely due to the lack of free hydroxyl groups in the RSV molecule. Therefore, our results may support the already observed correlation between the number of hydroxyl groups and antioxidant activity in polyphenols.

On the contrary, as shown in Figure 8, RSVTR proved to be a better anti-glycation agent than RSV, while a physical mixture of RSV and trolox in the same molar ratio of RSVTR showed a significantly lower glycation inhibition than RSV and RSVTR. According to Yeh et al. [52], many steps are involved in AGE production and anti-glycation may occur in every step leading to the synthesis of AGEs. In the initial phase, the Maillard reaction between reducing sugars and the terminal amino groups of proteins, nucleic acids, or phospholipids form unstable Schiff bases that can be easily oxidized generating free radicals. In the intermediate phase, the process of Amadori rearrangement results in the production of many carbonyl compounds that form isomers with the arginine and lysine residues of proteins, the so-called AGEs, in the last phase of the Maillard reaction. Glycation can be suppressed at the earliest stage by scavenging free radicals and lowering the production of carbonyl and dicarbonyl compounds. Additionally, methylglyoxal (MG) trapping, a primarily reactive dicarbonyl compound, may inhibit AGE production.

As RSVTR showed a lower antioxidant ability compared to RSV, its superior anti-glycation activity could not be attributed to the reduction of free radical production. As shown in Table 1, RSVTR physicochemical properties significantly differ from those of RSV. The greater TPSA value and the larger number of RBN and HBA of RSVTR compared to RSV could account for its higher anti-glycation activity allowing a better interaction between RSVTR and bovine serum albumin, which reduced the chance of the protein reaction with fructose, the reducing sugar used to perform the anti-glycation assay. In addition, the chemical stability of RSVTR at pH and temperature values similar to those used in the anti-glycation assay (pH 7.4, 37 °C) makes it unlikely that the observed anti-glycation activity could be attributed to RSVTR degradation products. Other authors postulated the MG trapping by polyphenols to explain the increased inhibition of glycation they achieved [52]. A similar mechanism could occur in RSVTR inhibition of the glycation process but the results of this study did not allow providing any evidence to support this hypothesis. Therefore, further studies have been planned to elucidate the actual mechanisms involved in the improved anti-glycation activity of RSVTR compared to RSV.

## 4. Conclusions

Resveratrol esterification with trolox, a strong antioxidant analogue of vitamin E, resulted in a derivative (RSVTR) whose physicochemical properties differed from those of resveratrol. In particular, the increased values of topographical polar surface area, number of hydrogen bond acceptors, and rotatable bonds number could make RSVTR interaction with albumin stronger than that of RSV. Although RSVTR showed lower antioxidant activity than RSV, its ability to inhibit the glycation process was significantly higher than its parent drug. Therefore, these results encourage further in vitro and in vivo investigations to explore the actual potential of RSVTR as an anti-glycation agent.

## Figures and Tables

**Figure 1 antioxidants-10-00012-f001:**
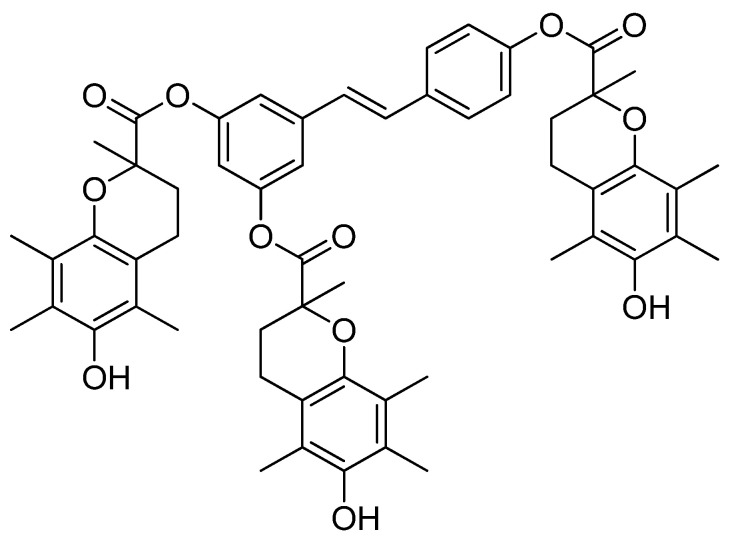
Resveratrol triester with trolox (RSVTR).

**Figure 2 antioxidants-10-00012-f002:**
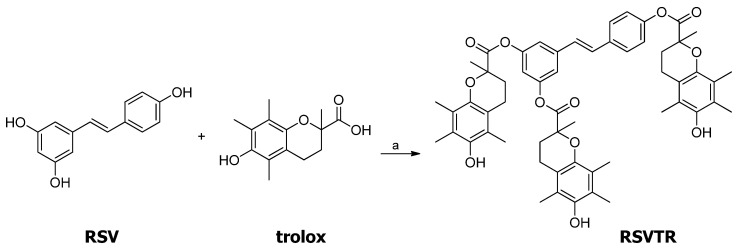
Synthesis of RSVTR. Reagents and conditions: (a) EDC · HCl, DMAP, dry DCM, 0 °C, then room temperature 8 h.

**Figure 3 antioxidants-10-00012-f003:**
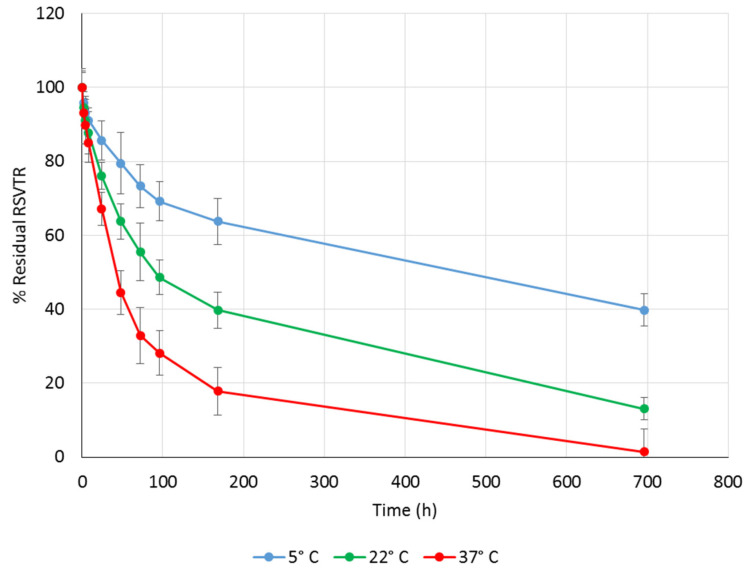
Percentage of residual RSVTR vs. time after incubation at pH 7.2 and different temperatures.

**Figure 4 antioxidants-10-00012-f004:**
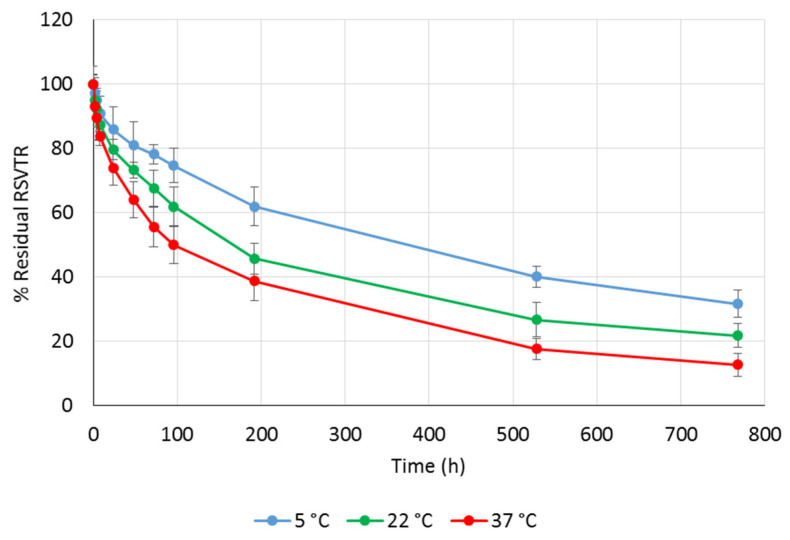
Percentage of residual RSVTR vs. time after incubation at pH 6.8 and different temperatures.

**Figure 5 antioxidants-10-00012-f005:**
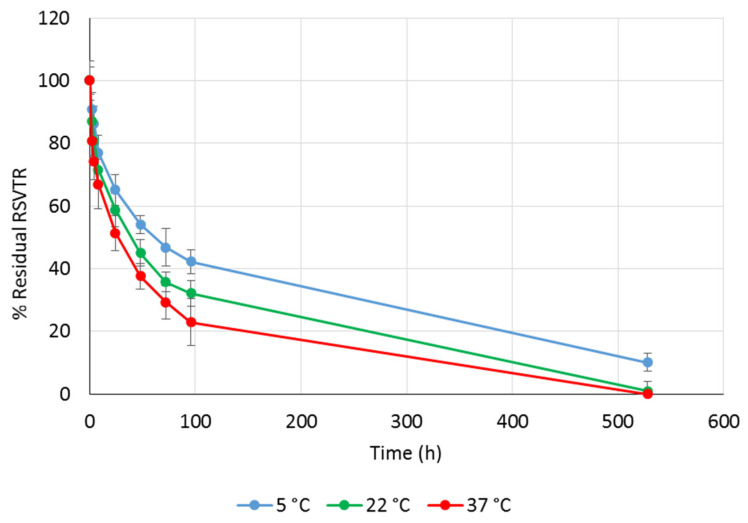
Percentage of residual RSVTR vs. time after incubation at pH 2.0 and different temperatures.

**Figure 6 antioxidants-10-00012-f006:**
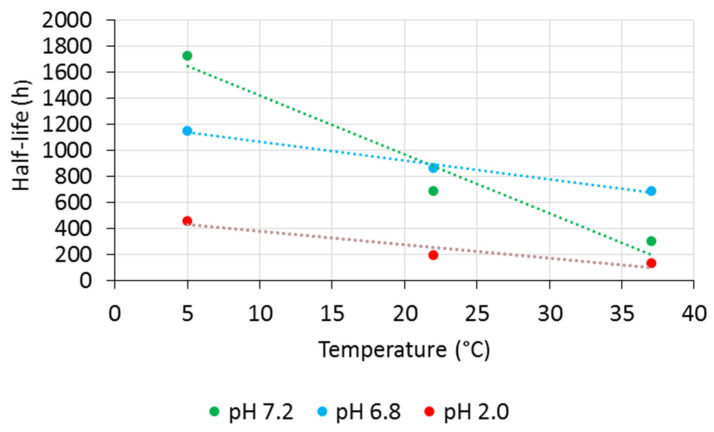
RSVTR half-life vs. temperature of incubation at different pH values.

**Figure 7 antioxidants-10-00012-f007:**
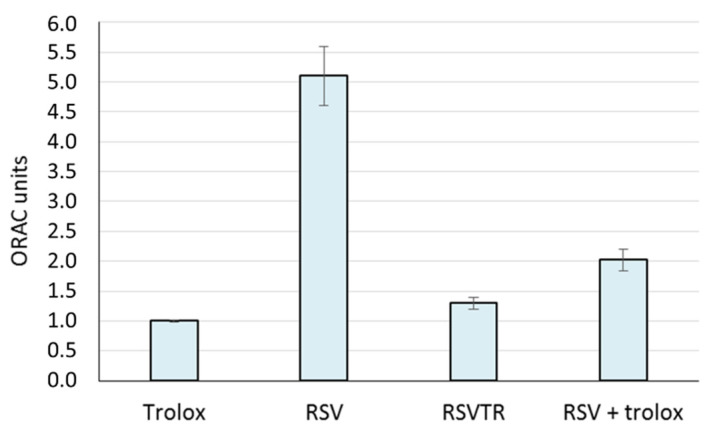
Oxygen Radical Absorbance Capacity (ORAC) units of trolox, resveratrol (RSV), resveratrol triester with trolox (RSVTR) and a physical mixture of RSV and trolox in the same molar ratio of RSVTR (RSV + trolox).

**Figure 8 antioxidants-10-00012-f008:**
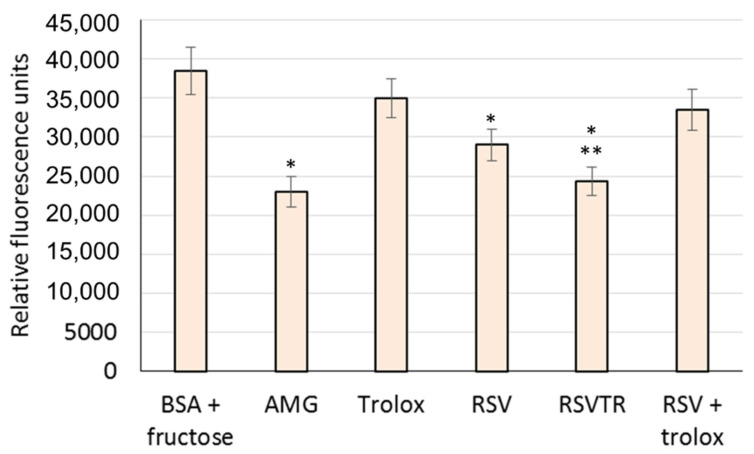
Relative fluorescence units of bovine serum albumin (BSA) incubated with fructose and in presence of aminoguanidine (AMG) as reference or in presence of trolox (positive control), resveratrol (RSV), resveratrol trimester with trolox (RSVTR) or a physical mixture of RSV and trolox in the same molar ratio of RSVTR (RSV + trolox). Statistical analysis for the comparison: * *p* < 0.05 vs. BSA + fructose; ** *p* < 0.05 vs. RSV.

**Table 1 antioxidants-10-00012-t001:** Physicochemical properties and molecular descriptors of resveratrol (RSV), trolox, and resveratrol triester with trolox (RSVTR).

Compound	Parameter ^1^									
	Formula	MW	mp °C	S_w_	cLogP	TPSA	HLB	HBD	HBA	RBN
RSV	C_14_H_12_O_3_	228.25	254	>6 mg	3.40	60.69	5.42	3	3	2
trolox	C_14_H_18_O_4_	250.29	187–189	>6 mg	3.66	69.59	6.94	2	4	1
RSVTR	C_56_H_60_O_12_	925.08	113–116	<1 mg	14.78	167.28	3.22	3	9	11

^1^ MW, molecular weight; mp, melting point; S_w_, water solubility in 100 mL; HLB, hydrophilic–lipophilic balance; cLogP, calculated LogP; TPSA, topological polar surface area; HBD, number of hydrogen bond donors; HBA, number of hydrogen bond acceptors; RBN, rotatable bonds number. Values calculated using Marvin 20.11.0, ChemAxon (https://www.chemaxon.com).

**Table 2 antioxidants-10-00012-t002:** Half-life (h) of resveratrol triester with trolox after incubation at different pH values and different temperatures.

pH	Temperature (°C)		
	5	22	37
2.0	462	198	133
6.8	1155	866	693
7.2	1732	693	301

## Data Availability

Data is contained within the article.

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
