# Peer review of "In Vitro Antioxidant and Anti-Glycation Activity of Resveratrol and Its Novel Triester with Trolox"

_antioxidants, 2020, doi:10.3390/antiox10010012_

Round 1

Reviewer 1 Report

Resveratrol is well known for its many beneficial activities, but its unfavorable physico-chemical properties impair its effectiveness after systemic and topical administration; thus, several strategies have been investigated to improve resveratrol efficacy. The aim of this study was to synthesize a novel resveratrol triester with Trolox and to evaluate its antioxidant and anti-glycation activity compared to resveratrol. 

Please consider the following suggestions:

  • Please use italic font when you mention in vivo and in vitro, in all the manuscript.
  • Page 6 – Table 1. Please replace Compd with Compound.
  • Rows 130–131 – please indicate the appropriate intervals of time and explain the reasons for choosing them.

Author Response

Resveratrol is well known for its many beneficial activities, but its unfavorable physico-chemical properties impair its effectiveness after systemic and topical administration; thus, several strategies have been investigated to improve resveratrol efficacy. The aim of this study was to synthesize a novel resveratrol triester with Trolox and to evaluate its antioxidant and anti-glycation activity compared to resveratrol.

Answer

We would like to thank the reviewer for reviewing our manuscript.

Please consider the following suggestions:

Please use italic font when you mention in vivo and in vitro, in all the manuscript.

Answer

As requested, we used italic font mentioning in vivo and in vitro throughout the manuscript.

Page 6 – Table 1. Please replace Compd with Compound.

Answer

As requested, we replaced the abbreviation Compd with Compound in Table 1. Additionally, we inserted the abbreviation Sw to replace “Water sol. 100 mL” and we explained the meaning of the abbreviation Sw in the footnote of Table 1.

Rows 130–131 – please indicate the appropriate intervals of time and explain the reasons for choosing them.

Answer

As requested, we indicated the intervals of time we used and we explained the reason for this choice. Therefore, we amended the text (line 131 revised version) as follows:

All solutions containing the same amount of RSVTR (1.18 µM) were kept in the dark at a constant temperature of 37 °C, 22 °C and 5 °C.  At appropriate time intervals, chosen to obtain enough data to pinpoint the stability profile (0, 2, 4, 8, 24, 48, 72, 96, 168, 696 h at pH 7.2; 0, 2, 4, 8, 24, 48, 72, 96, 192, 528, 768 h at pH 6.8; 0, 2, 4, 8, 24, 48, 72, 96, 528 h at pH 2.0), samples were withdrawn and analyzed immediately by HPLC for the residual RSVTR.

Reviewer 2 Report

In recent years, there has been an increasing body of evidence that resveratrol has a beneficial effect on the body and may be useful in the prevention and treatment of certain metabolic disorders, including diabetes. The positive effect of resveratrol in diabetic patients results from: 1) reduction of blood glucose concentration, 2) protection of pancreatic beta cells, 3) reduction of hyperinsulinemia and insulin resistance of tissues. There is a lot of evidence supporting the antihyperglycemic effect of the resveratrol. Therefore, the research undertaken in this work is very promising, but undoubtedly requires further analysis.

There is a need to explain why the RSVTR antioxidant capacity is so low. Perhaps it would be worth verifying this activity with other methods known from the literature. Why was this method chosen for the determination of antioxidant activity?

Certainly, the presented results can be treated as an introduction to further research, because the discussed problem is undoubtedly very important.

Author Response

In recent years, there has been an increasing body of evidence that resveratrol has a beneficial effect on the body and may be useful in the prevention and treatment of certain metabolic disorders, including diabetes. The positive effect of resveratrol in diabetic patients results from: 1) reduction of blood glucose concentration, 2) protection of pancreatic beta cells, 3) reduction of hyperinsulinemia and insulin resistance of tissues. There is a lot of evidence supporting the antihyperglycemic effect of the resveratrol. Therefore, the research undertaken in this work is very promising, but undoubtedly requires further analysis.

There is a need to explain why the RSVTR antioxidant capacity is so low. Perhaps it would be worth verifying this activity with other methods known from the literature. Why was this method chosen for the determination of antioxidant activity?

Certainly, the presented results can be treated as an introduction to further research, because the discussed problem is undoubtedly very important.

Answer

We would like to thank the reviewer for reviewing our manuscript and for the relevant comments.

As reported in literature (Amić et al., SAR and QSAR of the antioxidant activity of flavonoids. Curr Med Chem. 2007, 14, 827–845. DOI: 10.2174/092986707780090954; Takano-Ishikawa et al., Structure–activity relations of inhibitory effects of various flavonoids on lipopolysaccharide-induced prostaglandin E 2 production in rat peritoneal macrophages: Comparison between subclasses of flavonoids. Phytomedicine 2006, 13, 310–317. DOI: 10.1016/j.phymed.2005.01.016, Ronsisvalle et al., Natural flavones and flavonols: relationships among antioxidant activity, glycation and metalloproteinase inhibition. Cosmetics 2020, 7, 71; DOI: 10.3390/cosmetics7030071), the number of hydroxyl groups in the molecule of polyphenols strongly affects the anti-oxidant and anti-glycation activity of such compounds. In this work, we synthesized a derivative of resveratrol (a polyphenol) esterifying three hydroxyl groups of resveratrol with trolox. This derivative provided a lower antioxidant activity compared to resveratrol likely due to the lack of free hydroxyl groups in the resveratrol molecule. Therefore, our results may support the already reported correlation between number of hydroxyl groups and anti-oxidant activity in polyphenols.

Therefore, we added in the text the above-mentioned explanation (line 267, revised version) as follows:

As reported in the literature [46], the number of hydroxyl groups in polyphenols strongly affects the anti-oxidant and anti-glycation activity of such compounds. In this work, we synthesized a derivative of RSV (a polyphenol) esterifying three hydroxyl groups of RSV with trolox. The lower antioxidant activity of RSVTR compared to RSV could be likely due to the lack of free hydroxyl groups in the RSV molecule. Therefore, our results may support the already observed correlation between the number of hydroxyl groups and anti-oxidant activity in polyphenols.

As we have already used the oxygen radical absorbance capacity (ORAC) assay to compare the anti-oxidant activity of polyphenols (Ronsisvalle et al., Natural flavones and flavonols: relationships among antioxidant activity, glycation and metalloproteinase inhibition. Cosmetics 2020, 7, 71; DOI: 10.3390/cosmetics7030071), we chose to use the ORAC assay as it provided us with reliable results that were similar to those obtained by others using different types of assays such as the DPPH assay.

Furthermore, Schaich et al. (Hurdles and pitfalls in measuring antioxidant efficacy: A critical evaluation of ABTS, DPPH, and ORAC assays, Journal of Functional Foods, Volume 18, Part B, 2015, 782-796. DOI:10.1016/j.jff.2015.05.024) reported that “The ORAC assay offers several advantages over ABTS.+ and DPPH assay. It uses peroxyl radicals that are better models of antioxidant reactions with oxidizing lipids and reactive oxygen species (ROS) in foods and in vivo, and it provides continuous generation of radicals on a realistic time scale (more like actual reactions in situ).

Reviewer 3 Report

In this manuscript, the authors synthesized a novel RSV triester with trolox, an analogue of vitamin E. The new RSV derivative (RSVTR) was assayed in vitro to evaluate its antioxidant and anti-glycation activity compared to RSV. RSVTR chemical stability was assessed at pH 2.0, 6.8, and 7.2 and different storage temperatures (5, 22 and 37°C). RSVTR showed a lower antioxidant ability than RSV (determined by the oxygen radical absorbance capacity assay) while its anti-glycation activity (evaluated using the Maillard reaction) was significantly greater than that of RSV. However, I do not recommend accepting this manuscript because of the following issues: 1. Although the authors tested the stability of RSVTR through several experiments, they did not carry out enough biological activity tests to verify the advantages of this compound to prove that it has good in vitro antioxidant anti-glycation activity, which is more potent than RSV. More biological evaluations are needed. 2. When performing the biological evaluations, one equivalent of RSVTR should be compared to one equivalent together with one equivalent of Trolox. Only in this way, it is reasonable to compare the advantage effects of the designed compound.

Author Response

In this manuscript, the authors synthesized a novel RSV triester with trolox, an analogue of vitamin E. The new RSV derivative (RSVTR) was assayed in vitro to evaluate its antioxidant and anti-glycation activity compared to RSV. RSVTR chemical stability was assessed at pH 2.0, 6.8, and 7.2 and different storage temperatures (5, 22 and 37°C). RSVTR showed a lower antioxidant ability than RSV (determined by the oxygen radical absorbance capacity assay) while its anti-glycation activity (evaluated using the Maillard reaction) was significantly greater than that of RSV. However, I do not recommend accepting this manuscript because of the following issues: 1. Although the authors tested the stability of RSVTR through several experiments, they did not carry out enough biological activity tests to verify the advantages of this compound to prove that it has good in vitro antioxidant anti-glycation activity, which is more potent than RSV. More biological evaluations are needed. 2. When performing the biological evaluations, one equivalent of RSVTR should be compared to one equivalent together with one equivalent of Trolox. Only in this way, it is reasonable to compare the advantage effects of the designed compound.

Answer

We would like to thank the reviewer for reviewing our manuscript. As the reviewer noticed, in this manuscript, we performed several experiments to determine the stability in different experimental conditions of the new resveratrol derivative we synthesized. In our opinion, well-designed stability studies are essential to evaluate the “drug-likeness” of novel drug derivatives. Therefore, prior to performing extensive biological studies, we focused on determining the stability of the derivative under investigation. As we highlighted in “Results and Discussion” and in “Conclusions”, we have already planned further in vitro and in vivo biological studies to investigate more in-depth the activity of such derivative.

As reported in the manuscript, one mole of the derivative we synthesized consisted of one mole of resveratrol and three moles of trolox. Therefore, to demonstrate that the observed activity was due to the new molecule and it was not dependent on resveratrol and trolox as such, we compared the activity of one mole of derivative with a mixture of one mole of resverqtrol and three moles of trolox.

As suggested by the reviewer, further in vitro biological studies will be performed comparing one equivalent of derivative to one equivalent of resveratrol together with one equivalent of trolox.

Reviewer 4 Report

The authors demonstrate that the new compound, a derivative of resveratrol and trolox, has less antioxidant capacity, but more anti-glycation capacity than resveratrol. 

The Results and Discussion sections are combined. I think they should be formally separated.

The Conclusions are good. More in vitro and in vivo studies are needed to determine if this compound is worth pursuing. 

There are some typos/grammatical errors. 

Author Response

The authors demonstrate that the new compound, a derivative of resveratrol and trolox, has less antioxidant capacity, but more anti-glycation capacity than resveratrol.

The Results and Discussion sections are combined. I think they should be formally separated.

Answer

We would like to thank the reviewer for reviewing our manuscript.

According to the “instructions for authors” of the journal Antioxidants, Discussion may be combined with Results. We found more convenient to combine Results and Discussion to illustrate our data as concisely as possible avoiding useless repetitions.

The Conclusions are good. More in vitro and in vivo studies are needed to determine if this compound is worth pursuing.

Answer

We agree with the reviewer that more in vitro and in vivo studies are needed to determine the usefulness of the new resveratrol derivative we synthesized and we highlighted this aspect in the “Conclusions”.

There are some typos/grammatical errors.

Answer

We apologize for all typos/grammatical errors. We amended the text to eliminate such mistakes.

Round 2

Reviewer 3 Report

Although several issues pointed by the other reviewers have been addressed, additional biological evaluation is still needed to be performed.